# Functionally Relevant Cytokine/Receptor Axes in Myelofibrosis

**DOI:** 10.3390/biomedicines11092462

**Published:** 2023-09-05

**Authors:** Giulia Pozzi, Cecilia Carubbi, Giacomo Maria Cerreto, Chiara Scacchi, Samuele Cortellazzi, Marco Vitale, Elena Masselli

**Affiliations:** 1Anatomy Unit, Department of Medicine & Surgery (DiMeC), University of Parma, 43126 Parma, Italy; 2University Hospital of Parma, AOU-PR, 43126 Parma, Italy

**Keywords:** inflammation, myeloproliferative neoplasms, myelofibrosis, cytokine receptors, CD34^+^ cells, hematopoietic niche anatomy

## Abstract

Dysregulated inflammatory signaling is a key feature of myeloproliferative neoplasms (MPNs), most notably of myelofibrosis (MF). Indeed, MF is considered the prototype of onco-inflammatory hematologic cancers. While increased levels of circulatory and bone marrow cytokines are a well-established feature of all MPNs, a very recent body of literature is intriguingly pinpointing the selective overexpression of cytokine receptors by MF hematopoietic stem and progenitor cells (HSPCs), which, by contrast, are nearly absent or scarcely expressed in essential thrombocythemia (ET) or polycythemia vera (PV) cells. This new evidence suggests that MF CD34^+^ cells are uniquely capable of sensing inflammation, and that activation of specific cytokine signaling axes may contribute to the peculiar aggressive phenotype and biological behavior of this disorder. In this review, we will cover the main cytokine systems peculiarly activated in MF and how cytokine receptor targeting is shaping a novel therapeutic avenue in this disease.

## 1. Introduction

Myelofibrosis (MF) is a chronic, potentially life-threatening hematologic neoplasm that includes primary MF (PMF), post-essential thrombocythemia MF (post-ET-MF), and post-polycythemia vera MF (post-PV-MF). MF is a heterogeneous disorder, ranging from a mild disease phenotypically mimicking ET, as in the case of a pre-fibrotic MF (pre-PMF), to an overt disease (overt-PMF and post-ET/post-PV-MF) typified by ineffective clonal hematopoiesis, extramedullary hematopoiesis, bone marrow stromal changes resulting in reticulin deposition and fibrosis, and a propensity for leukemia transformation [1]. Mutations in *JAK2*, *CALR*, and *MPL* genes occur in the great majority of MF patients, underscoring the role of constitutive JAK/STAT activation in disease initiation and maintenance.

However, the phenotypic and prognostic heterogeneity of MF clearly suggests that additional biological factors contribute to the pathophysiology of this disease. Specifically, chronic aberrant pro-inflammatory cytokine signaling is considered a driving force for the development of end-stage disease in MPNs, i.e., MF [2].

Cancer-related inflammation represents the seventh hallmark in the development of cancer, enabling unlimited replicative potential, independence from growth factors, resistance to inhibitory signals, escape of programmed cell death, enhanced angiogenesis, tumor extravasation, and eventually metastasis [3]. Cancer cells hijack inflammatory mechanisms to promote their own growth and survival, by releasing inflammatory mediators that alter tissue homeostasis at both the local and systemic level. This inflammatory microenvironment, characterized by persistently activated immune cells, increased cytokines, accumulation of ROS and consequent tissue damage/remodeling, in turn elicits the expansion, selection and evolution of the malignant clone, therefore generating a vicious cycle that leads to cancer establishment and progression.

In addition, there is evidence supporting the notion that chronic inflammatory conditions—ranging from infection-mediated chronic inflammation, autoimmune disorders, obesity, inflammaging, etc.—favors carcinogenesis by creating a permissive environment for the onset and expansion of the malignant clone [2,4,5,6]. This has also been demonstrated in the case of MPNs, in which several studies have shown that inflammatory diseases may precede the diagnosis of ET, PV, and MF [7].

## 2. The “Cytokine Storm” in MF: Soluble Cytokines

In MPNs, the chronic inflammatory state generated by mutant hematopoietic stem/progenitor cells promotes bone marrow (BM) mesenchymal stromal cell reprogramming into fibrosis-driving myofibroblasts [8,9,10]. BM fibrosis, in turn, leads to the progressive disruption of the anatomy and function of bone marrow niche into a disease-permissive environment. Indeed, an overtly fibrotic stage, such as in the case of overt-PMF and PPV/PET-MF, dictates a more severe disease stage with dismal prognosis and higher risk of leukemic evolution [11]. Therefore, many efforts have been undertaken to identify cytokines and chemokines involved in MF development, progression, and outcome. Table 1 summarizes landmark studies assessing levels of cytokines/chemokines and growth factors in MF as compared to healthy subjects or other MPNs. In Table 2, we focused on the inflammatory mediators having an impact on MF phenotype. General cytokine/chemokine/growth immunological function is reported, followed by its specific effect(s) on disease characteristics, association with driver mutational status, and outcome.

Overall, an increase in both pro-inflammatory (IL-1α and β, IL-2, sIL-2R, IL-6, IL-12, IL-15, IL-17, INF-α and γ, and TNFα) and anti-inflammatory (IL-4, IL-10, and IL-13) cytokines, chemokines (IL-8, IP-10/CXCL-10, RANTES, MCP-1/CCL2, MIG, and MIP-1α and β), and growth factors (TGF-β, G-CSF, HGF, and VEGF) is detected among studies. Higher levels of IL-1α and β, IL-6, IL-8, IL-13, MCP-1/CCL2, and TGF-β correlate with a higher degree of bone marrow fibrosis. HGF, IL-6, IL-8, and IP-10/CXCL-10 are associated with the presence of *JAK2*V617F mutation and a proliferative phenotype (leukocytosis and hepatosplenomegaly). IL-2, sIL-2R, IL-6, IL-8, IL-12, IL-17, MCP-1/CCL2, and MIP-1α correlate with anemia/RBC transfusion dependency while IL-6, IP-10/CXCL-10 and MIP-1β correlate with thrombocytopenia.

The relevance of the pro-inflammatory milieu in MF is further underscored by the fact that increased levels of IL-8, IL-2R, IL-12, IL-15, MIP-1α and IP-10 are independently predictive of shorter overall survival. However, when including the revised Dynamic International Prognostic Scoring System (DIPSS-plus) in multivariate analyses, only IL-8, IL-2R, IL-12, and IL-15 retain their prognostic value. Of note, among them, IL-8 also correlated with reduced leukemia-free survival (LFS).

Some of the cytokines increased in MF play an important role in inflammasome biology, such as IL-1, TNFα, IL-6, and INFs.

Inflammasomes are a class of cytosolic multiprotein complexes that promote the expression, maturation, and release of pro-inflammatory cytokines to engage innate immune defenses [28]. 

Nucleotide binding domain (NOD)-like receptor protein 3 (NLRP3) inflammasome is the best-studied of these multiprotein complexes. It is expressed in several cells including monocytes, lymphocytes, endothelial cells, HSPCs, and contributes to cancer pathophysiology by excessive production of cytokines [28,29,30]. NLRP3 inflammasome consists of NLRP3 protein, ASC, and procaspase 1. Cellular expression of NLRP3 is induced, in an NF-kB-dependent manner, by the so-called “signal 1” that includes liposaccharide (LPS) and members of the senescence-associated secretory phenotype (SASP), such as TNF-α and IL-6. Subsequently, activation of NLRP3 is mediated by the so-called “signal 2”, which is represented by damage- or pathogen-related signals (DAMP or PAMP) or by glucose or amino acid efflux. NLRP3 activation finally induces the NLRP3 protein/ASC/procaspase-1 inflammasome assembly and thus the autoproteolytic cleavage of procaspase-1 into active caspase-1. Caspase-1 converts pro-IL-1β and pro-IL-18 into their active forms, eventually leading to pyroptotic cell death and cytokines secretion [31]. 

Therefore, it has been suggested that NLRP3 inflammasome may be involved in MPN pathophysiology promoting the cytokine storm that characterizes these disorders. However, data on this topic are still scant and only very recently Zhou et al. provided the first evidence of a higher NLRP3 inflammasome-related gene signature in MPNs. The authors analyzed expression levels and genetic polymorphisms of inflammasome genes *NLRP3*, *NF-KB*, *CARD8*, *IL1B*, and *IL18* on BM cells of MPN patients, demonstrating that an increased expression of NLRP3 inflammasome-related genes was associated with *JAK2*V617F mutation, WBC counts, and splenomegaly [32]. Despite this, the functional effects of inflammasome pathways in MPN pathogenesis are currently unknown. 

## 3. Cytokine/Chemokine Axes Functionally Relevant in MF

Despite the robust body of literature now available on the cytokine profile of MPN patients (recently reviewed by our group [33]), only a few studies approached the functional relevance of the interplay between cytokines and their receptors. In this section we will examine the studies that, so far, demonstrated the mechanistic implications of specific cytokine/chemokine axes in MF pathogenesis. 

### 3.1. IL-1/IL-1R Axis

IL-1 was the first cytokine to be discovered and is the most important member of the IL-1 family, which includes both pro-inflammatory and anti-inflammatory mediators. The IL-1 family consists of 11 members: IL-1β, IL-1α, IL-18, IL-33, IL-1F5 to IL-1F10, and IL-1 receptor antagonist (IL-1RA). IL-1α and IL-1β (collectively termed IL-1) are pro-inflammatory cytokines of about 17–18 kDa, with a well-established role in innate immunity, inflammatory response, and hemopoiesis [34,35,36,37,38,39]. On the other side, IL-1RA is a natural competitive inhibitor of IL-1β and IL-1α that binds to the IL-1 receptor, hampering agonist binding and thus preventing cytokine-dependent signal transduction. 

Nine members of the IL-1 family, including IL-1α and IL-1β, occur in a single cluster on the long arm of human chromosome 2 (at position 2q13-2q21) and likely derive from a series of gene duplications of the prototypical cytokine IL-1β [35]. IL-1β is mainly produced by cells of the myelomonocytic lineage, including monocytes, macrophages, and dendritic cells, upon microbial invasion and tissue injury. IL-1β, along with all other IL-1 family cytokines (except for IL-1RA), is initially produced as a biologically inactive precursor, called pro-IL-1β, requiring cleavage by caspase-1 to be converted into its active form. Therefore, inflammasome is a key driver of its activation as described above [40]. IL-1α, conversely, is not a typical inflammatory cytokine as its precursor protein (pro-IL-1α) is constitutively present in a wide variety of cells, such as in macrophages and in non-immune stromal cells (fibroblasts, epithelial cells, and endothelial cells) where it acts as a transcription factor, regulating cellular functions, proliferation, senescence, and apoptosis [41,42]. Inflammatory stimuli induce the cleavage of pro-IL-1α in two domains: the C-terminal domain acts as an extracellular cytokine triggering pro-inflammatory cascades, while the intracellular precursor domain acts as transcription factor upregulating the expression of inflammation-related genes. For these reasons, unlike IL-1β, IL-1α is considered a ‘dual function’ cytokine [42,43].

Upon secretion, both IL-1α and IL-1β bind to the same receptors, the IL-1R. Additionally, IL-1α can also be exposed on the cell membrane of activated macrophages for juxtacrine signaling [44].

IL-1 receptors, namely IL-1R1 and IL-1R3 (also known as IL-1R accessory protein, IL-1RAP or IL-1RAcP), are integral membrane proteins forming an active trimeric complex binding IL-1α and β. This complex rapidly assembles intracellular signaling proteins, including MyD88 [35], IRAK, and TRAF [39], that in turn activate the NF-κB, c-Jun N-terminal kinase (JNK), and p38 mitogen-activated protein kinase (MAPK) pathways involved in several cellular processes such as proliferation, differentiation, and apoptosis [45,46,47,48].

IL-1β is considered a master regulator of inflammation that induces the production of ROS and NO and that controls the production of chemokines (as CXCL8, CCL2) and inflammatory cytokines (as TNF-α and IL-6) and its own expression via a positive feedback loop in an autocrine or paracrine manner. Several associations have been established between upregulated IL-1 signaling and tumorigenic mechanisms, including inflammation-induced carcinogenesis, invasiveness, metastasis, and angiogenesis [37]. Additionally, there is evidence of a role for IL-1β in conditions associated with organ (lung, liver, skin, or kidney) fibrosis [49].

In MPNs, circulating levels of IL-lα and IL-1β have been assessed in several studies with contradictory results. Former investigations demonstrated that in all three MPN, serum levels of IL-1α and IL-1β were comparable to healthy subjects and nearly undetectable during the stable phase of the disease [13,16].

Conversely, in more recent studies, plasma levels of IL-1β and IL-1RA have been found to be significantly higher in PMF patients as compared to healthy controls and PV patients [20]. Furthermore, in MF patients (both primary and secondary), elevated IL-1RA levels predicted poor anemia response and correlated with marked splenomegaly [12,16,17]. Additionally, by comparing the cytokine gene expression profile of bone marrow cells using Nanostring technologies, Wong et al. showed that the RNA transcript of *IL1B*, along with other pro-inflammatory cytokines, was 1.5-fold higher in overt-PMF as compared to pre-PMF patients, suggesting the association of *IL1B* gene expression with bone marrow fibrosis [50].

In 2022, two different research groups simultaneously described the functional implication of IL-1/IL-1R1 axis activation in MPNs in general, and MF in particular [14,15]. The authors found that IL-1β, IL-1RA [14], and IL-1α [15] were elevated in *JAK2*V617F-positve MPNs as compared to healthy controls, with PV and PMF subjects displaying higher levels than ET [14]. Moreover, cytokine transcript levels in peripheral blood granulocytes correlated with *JAK2*V617F allele burden [14,15]. Of note, Rai et al. assessed the expression of IL-1 receptor by MPNs vs. control HSCs, showing a 3-fold increase in the frequency of IL-R1+ and IL-1RAcP+ circulating HSCs in MPNs as compared to healthy subjects. As described for the IL-1β transcript, the percentage of IL-1R1+ and IL-1RAcP + cells correlated with *JAK2*V617F allele burden as well. Notably, HSCs from PMF patients, besides overexpressing IL-1 receptor, show a higher expression of IL-1R pathway target genes [14]. Both genetic knockout and pharmacological inhibition (by blocking antibodies) of IL-1β ameliorates the fibrotic phenotype of *JAK2*V617F MPN mice, the latter treatment also synergizing with ruxolitinib [14].

In line with these findings, Rahman et al. showed that (i) exogenous administration of IL-1β in heterozygous *JAK2*V617F knock-in mice, that mainly exhibit a PV-like disease phenotype, fostered disease progression (granulocyte and megakaryocyte hyperplasia coupled with impaired erythropoiesis, and increased reticulin fibers); (ii) IL-1R1 blockade by anti-IL1R1 antibodies was capable of significantly reducing the disease stigmata in a homozygous *JAK2*V617F knock-in mouse model of MF (normalized blood cell counts, reduced spleen size, BM fibrosis, and abnormal MK clustering) [15]. 

These effects on the megakaryocytic lineage suggest that the mechanisms by which IL-1 promotes myelofibrosis involve direct effects on MKs. MKs are well-established key players in MF pathogenesis [51,52,53,54,55,56] and ex vivo stimulation of BM cells from *JAK2*V617F mice with IL-1α or IL-1β, increased CFU-MK formation, and proliferation of primary MKs [17]. IL-1 signaling also affects normal BM mesenchymal stromal cells (MSC) and it may therefore play a role in the detrimental activation of the non-clonal stromal counterpart in MPNs. In fact, upon stimulation of BM MSC with IL-1β, a significant upregulation of *IL1A*, *IL1B*, *IL6*, *CCL2*, *CCL5*, and collagen gene transcripts takes place, therefore sustaining BM fibrotic transformation [17].

### 3.2. CXCL8/CXCR1/2 Axis

IL-8, also known as C-X-C motif chemokine ligand 8 (CXCL8), is initially known as a CXC chemokine recruiting neutrophils into areas of inflammation, infection, or injury [57] and therefore driving the acute inflammatory response. The gene that encodes for CXCL8 is located on chromosome 4q12-q21 [58]. *CXCL8* is initially translated as a 99-amino acid protein and it is subsequently processed into its active isoforms, which are either a 77-amino acid peptide in non-immune cells or a 72-amino acid peptide in monocytes and macrophages [59].

CXCL8 is secreted by a variety of cells, including monocytes, neutrophils, epithelial cells, fibroblasts, endothelial cells, mesothelial cells, macrophages, and airway smooth muscle cells, upon inflammatory stimuli, while it is usually undetectable in unstimulated cells because of the repression of the promoter [60]. Activating stimuli capable of causing a 5–100-fold increase in *CXCL8* expression include IL-1, TNFα, viral or bacterial products, other environmental stresses, and transcription factors (including activator protein-1 (AP-1) and NF-κB). Maximal CXCL8 expression and secretion can be due to the following molecular events: (1) de-repression of *CXCL8* gene promoter; (2) transcriptional activation of the *CXCL8* gene by NF-κB and JNK pathways; and (3) stabilization of *CXCL8* mRNA by the p38 MAPK pathway [60,61].

CXCL8 binding receptors include two G-protein-coupled receptors: CXCR1 (Interleukin-8 receptor A, IL-8RA) and CXCR2 (Interleukin-8 receptor B, IL-8RB), which share 78% sequence homology. CXCR1 and CXCR2 are expressed mainly on monocytes, granulocytes, and endothelial cells [62]. Although CXCL-8 dimer and monomer have identical affinities for CXCR2, CXCL8 monomer preferentially binds to CXCR1 [60]. Upon CXCL8 binding, the following stable complexes will be formed: CXCL8 (monomer)-CXCR1/2-G protein and CXCL8 (dimer)-CXCR2-G protein. These complexes activate multiple G-protein-mediated signaling cascades, namely: (i) phosphatidyl-inositol-3-kinase (PI3K)/Akt pathway; (ii) phospholipase C/protein Kinase C pathway; and (iii) MAPK pathway, triggering the phosphorylation of protein tyrosine kinases, including FAK and Src kinases [63,64,65]. Of note, among transcription factors activated by CXCR1/2 engagement, NF-κB plays a major role in fostering CXCL8 secretion in an autocrine loop.

CXCL8 is overexpressed in several types of solid cancers, with the serum level of CXCL8 acting as a prognostic marker [66,67]. Neoplastic cells are able to produce and release CXCL8, which in turn acts on the tumor microenvironment, promoting angiogenesis, epithelial–mesenchymal transition, increased endothelial cell permeability, suppressing anti-tumor immunity, and eventually favoring clone motility and invasiveness. In addition, CXCL8 acts in an autocrine manner, promoting tumor cell growth and survival [66].

In MPNs, as described above (see also Table 2), data from Ayalew Tefferi’s laboratory demonstrated that higher levels of circulating CXCL8 in (P)MF were an independent predictive factor of reduced survival on multivariate analysis, which included risk stratification according to the DIPSS plus as a covariate; moreover, they were correlated with leukocytosis, anemia, constitutional symptoms, increased circulating blasts, and a higher risk of leukemic transformation [16,17].

The functional role of the CXCL8/CXCR1/2 axis in MF has been recently elucidated by Ross Levine’s group [23]. Single-cell gene expression profile (scRNA-seq) of CD34^+^ cells isolated from MPN patients with different degrees of BM fibrosis, revealed that—regardless of the MPN subtype—overt-fibrotic patients were clustered together and enriched in genes involved in pro-inflammatory pathways (*TNFα*, *NF-KB*, *MIP2a*, and *MIP2b*), including CXCL8. Single-cell cytokine-secretome analysis revealed that MF patients showed the highest percentage of CXCL8-secreting CD34^+^ cells (54%) compared to other MPNs (31% in PV and 0% in ET). Of note, in these patients, the percentage of CXCL8-secreting CD34^+^ cells correlated with the degree of reticulin fibrosis and leukocytosis, hinting that CXCL8 secretion may serve as a biomarker for the presence of significant bone marrow fibrosis and a more aggressive disease. 

In vitro primary CD34^+^ colony formation capacity and the impact of CXCL8 treatment on proliferation were also assessed, showing that the presence of CXCL8 promoted the formation of CD34^+^ colonies, whereas the presence of CXCR1/2 inhibitors reversed this effect. Finally, in vitro cultured MF MK displayed increased CXCL8 secretion.

Focusing on the CXCL8 receptor, the authors showed that the percentage of CD34^+^ cells expressing CXCL8 receptor CXCR1 and 2 was significantly higher in MF than in healthy controls, indicating that MF hematopoietic progenitors could be both the source and the target of IL-8. Moreover, the administration of exogenous CXCL8 to neoplastic CD34^+^ cells from MF induced a marked cell proliferation, and a significant increase in CD33^+^ monocytic and CD41^+^ MK cell numbers, suggesting that the activation of IL-8/CXCR1/2 contributes to the creation of a self-fueling circle in MF [23].

Indeed, deletion of the *CXCR2* gene in the h*MPL*W515L adoptive transfer murine model of MF (Cxcr2^–/–^ h*MPL*W515) ameliorated blood cell parameters, decreased circulating pro-inflammatory cytokines, reduced MK BM infiltration, and led to a significant improvement of BM and spleen, resulting in increased overall survival in mice. Consistently, pharmacological inhibition of CXCR1/2 with reparixin improved hematological parameters and reduced BM fibrosis in the h*MPL*W515L adoptive transfer murine model of MF, synergizing with ruxolitinib [23]. 

The beneficial effects of CXCL8/CXCR1/2 axis blockage have been further documented by Annarita Migliaccio’s group in the *Gata1^low^* mouse model of MF. Along with enrichment in TGF-β1, BM MKs of *Gata1^low^* mice displayed higher levels of mCXCL1 (the murine equivalent of hCXCL8) and its receptors CXCR1 and CXCR2. According to these results, reparixin-treated mice showed a statistically significant decrease in BM fibrosis, and the degree of response was inversely correlated with reparixin plasma levels. Pharmacological treatment also proved to reduce TGF-β1 levels in Gata1*^low^* mice’s BM by restoring GATA-1 and reducing collagen III content in MKs, suggesting that CXCR1/2 blockage in Gata1^low^ mice ameliorates fibrosis by reducing TGF-β1 [25].

Overall, activation of the CXCL8/CXCR1/2 axis contributes to a permissive tumor microenvironment for the expansion of neoplastic clones, creating a self-fueling circle in MF. Indeed, neoplastic CD34^+^ cells are both the source and the target of CCL8, as they produce and secrete CXCL8 and express the receptor for this chemokine. The secretion of CXCL8 represents a biomarker of bone marrow fibrosis. The promising data obtained in mouse models of MF pave the way for future preclinical studies in the use of inhibitors of the CXCL8/CXCR1/2 axis in the therapy of MF.

### 3.3. CCL2/CCR2 Axis

The C–C motif chemokine ligand 2 (CCL2), also known as MCP-1 (monocyte chemotactic protein 1) is a member of the C-C chemokine family, which includes CCL7/MCP-3, CCL8/MCP-2, and CCL13/MCP-4 isoforms. Like the other members of the β chemokine family, the gene encoding for *CCL2* is located in the q11.2-12 cytoband of chromosome 17. CCL2 is typically secreted in two predominant forms (of 9 and 13 kDa) as a result of different O-glycosylation processes, which, however, do not affect their monocyte chemoattractant function. The N-terminal region is required for chemoattractant activity and is involved in protein dimerization. Usually, CCL2 binds its receptor as dimer, but there is some evidence that supports a model in which CCL2 binds and activates CCR2 in a monomeric form as well [68,69].

Although CCL2 is produced by a variety of cells, mononuclear cells such as monocyte/macrophages are considered the main source of this chemokine [70]. CCL2 production is triggered by growth factors, cytokines, oxidative stress and, similarly to IL-8, by pro-inflammatory mediators such as TNFα, IL-1β, IFN-γ, and PDGF [68,69]. NF-κB appears to be the main transcription factor, along with AP-1, involved in the transcription of the *CCL2* gene [71,72]. 

CCL2 exerts its biological functions by preferentially engaging the G-protein-coupled seven-transmembrane receptor CCR2. CCR2 has two different isoforms generated by alternative splicing, namely CCR2A and CCR2B [73]. 

The CCL2/CCR2 interaction induces the activation of a complex, intracellular signaling network which includes the phosphatidylinositol 3-kinase (PI3K)/AKT, p38 MAPK, and Janus kinase (JAK)/STAT3 pathways [74]. Through these pathways, it promotes cell proliferation and functional polarization (i.e., towards M1 monocyte/macrophage phenotype) and the cytoskeletal (re)-organization responsible of leukocyte chemiotaxis, cell migration, and extravasation [75].

Upregulation of the CCL2/CCR2 axis has been observed in various chronic inflammatory diseases displaying fibrosis as end-organ damage [76,77,78]. Moreover, CCL2 has been described as a “tumor-derived chemotactic factor” since it has been found frequently overexpressed in both neoplastic cells and tumor microenvironment stromal cells. In several studies, high CCL2 levels have been associated with a more aggressive cancer phenotype, a higher risk of metastasis, and poor prognosis [79,80] in a wide range of solid cancers including breast [81], prostate [82], colorectal [83], liver [84], and pancreatic cancers [85]. 

In hematologic neoplasms, CCL2/CCR2 axis has been investigated in Acute Myeloid Leukemia (AML) [86], multiple myeloma [87], and systemic mastocytosis [88]. Of note, in the latter case, Greiner and colleagues demonstrated that CCL2 overexpression in systemic mastocytosis was triggered by a somatic mutation of the *KIT* gene (*KIT*D816V) and that CCL2/CCR2 system activation was responsible for BM microenvironment alterations, including deposition of collagen fibers [88].

CCL2 is overexpressed in MPN patients and, when comparing single MPN entities, CCL2 levels tend to be higher in (P)MF as compared to ET or PV and in ET as compared to PV [33]. As discussed above, higher CCL2 levels correlated with transfusion dependency, marked splenomegaly, and significantly lower anemia response to IMIDs [16,17], clearly indicating a more aggressive disease.

In 2019, Wong et al. tested the expression of inflammatory genes by using Nanostring technology to directly measure RNA transcripts in BM biopsies of 108 MPNs, and they found a different gene expression profile in MPNs with fibrosis grade 0–1 (pre-fibrotic) vs. grade 2–3 (overtly fibrotic). Among differentially regulated genes, *CCL2* was up-regulated more than three-fold in overtly fibrotic as compared to pre-fibrotic MPNs [50]. 

One of the mechanisms that may underlie cyto/chemokine overexpression, is the presence of transcriptionally relevant single nucleotide polymorphism(s) (SNPs) [89]. Indeed, cyto/chemokine genes are known to be highly polymorphic. With respect to *CCL2*, the rs1024611 A-to-G SNP accounts for a sustained transcriptional activity of the gene leading to increased chemokine production [90]. We asked whether the rs1024611 genotype could affect the disease phenotype in MPNs, finding that the presence of the G-allele was associated with post-PV/ET-MF and a more aggressive disease phenotype in both PMF and post-PV/ET-MF patients (circulating blasts, lower hemoglobin, and higher grade of bone marrow fibrosis) [91]. Moreover, homozygosity for the rs1024611 SNP of *CCL2*, also responsible for higher chemokine production in this setting, was associated with reduced survival in PMF in both univariate and multivariate analysis including parameters of the IPSS scoring system [92].

The expression of the CCL2 binding counterpart (CCR2) was recently investigated in different hematopoietic cell types of MF patients. Barone and colleagues studied CCR2 expression on monocytes, typical target of this chemokine. They found that MF patients were enriched with monocytes expressing CCR2. Particularly, monocytes isolated from *JAK2*V617F-positive patients showed an altered chemokine receptor profile, characterized by an unusual CCR2 overexpression by intermediate and nonclassical monocytes vs. classic monocytes, that are the subset typically identified by CCR2 expression [93]. These data suggest an alteration of monocyte subsets and chemokine receptor profile in MF patients. 

Our laboratory first investigated the expression of CCR2 on hematopoietic progenitor cells from MPN patients. We demonstrated that only CD34^+^ cells from PMF express CCR2, which, by contrast, was virtually absent in healthy controls, PV, and ET. Activation of CCL2/CCR2 axis upon ex vivo CCL2 stimulation of CD34^+^CCR2^+^ PMF cells switched on pro-survival signals primarily mediated by the Akt pathway. By contrast, phosphorylation and activation of Akt or other pro-proliferative signal transduction pathways were absent in CD34^+^ cells from control subjects, which do not express CCR2 [92].

Additionally, we demonstrated that not only was the percentage of CD34^+^CCR2^+^ cells positively correlated with the degree of BM fibrosis in MPNs (a specular finding to what was described by Dunbar et al. for CXCR1/2) but it also proved an accurate diagnostic tool to discriminate among MPN subtypes with different degrees of bone marrow fibrosis. Indeed, CD34^+^CCR2^+^ flow cytometry detection in peripheral blood and/or bone marrow aspirates demonstrated a very good diagnostic performance in discriminating pre-PMF vs. true ET patients, as well as pre-PMF vs. overt-PMF patients, and also for patient longitudinal follow-up. Of note, focusing on MF patients (pre-PMF + overt-PMF + secondary MF), the percentage of CCR2-expressing CD34^+^ cells was enriched in intermediate-2/high vs. low/intermediate-1 risk patients (considering the composite risk scores IPSS/DIPSS for PMF and MYSEC-PM for sMF) and in patients with (≥1%) vs. without (<1%) circulating blasts, therefore correlating with a high-risk disease [94].

Jak-inhibitors may play an important role in dampening activated inflammatory signaling. Specifically, we demonstrated that the expression of CCL2 by circulating mononuclear cells and of CCR2 by circulating CD34^+^ cells was significantly reduced in patients under ruxolitinib treatment [92].

Overall, the selective activation of CCL2/CCR2 chemokine axis in MF may, on one side, boost clonal expansion and, on the other side, favor stromal changes in the BM; therefore, the use of CCR2 antagonists or CCL2 inhibitors could represent an attractive therapeutic strategy for this disease. 

### 3.4. IL4/IL13 Axis

IL-4 and IL-13 are related cytokines involved in type II inflammatory response, regulating many aspects of immune response triggered by pathogens and allergens. Both are members of the short-chain four helix bundle cytokine family which also includes IL-3, IL-5, and GM-CSF. They present approximately 30% structure homology and, since they share a common receptor chain, they trigger similar biological response [95].

The *IL4* and *IL13* genes lie only 13 kb apart within the same 160-kb region on the long arm of human chromosome 5, bands q23-31—the so-called Th2 cytokine locus—also containing the *IL5* gene. The *IL4*, *IL13*, and *IL5* genes are therefore coordinately regulated: upon cell stimulation, the LCR locus becomes accessible to transcription factors such as the NFAT proteins [96].

IL-4 and IL-13 are produced by CD4^+^ T cells, natural killer cells, type 2 innate lymphoid cells, mast cells, basophils, and eosinophils and regulate the responses of lymphocytes, myeloid cells, and non-hematopoietic cells such as airway smooth muscle cells [97]. 

The IL-13 and IL-4 receptor is a heterodimeric complex, composed of the IL-13Rα1 chain and the IL-4Rα chain [98]. The receptor chains lack endogenous kinase activity and thus utilize receptor-associated kinases, such as the Janus family tyrosine kinases (JAKs) and others, for signal transduction. The main signaling pathways activated by IL-13Rα1 and IL-4Rα chain heterodimerization are the JAK/STAT pathway, the IRS pathway, and the SHP-1/SHP-2/SHIP pathway [99]. Among them, STAT6 is the major transcription factor pathway activated in response to IL-4 and IL-13 binding, and STAT6 phosphorylation is crucial for the expression of IL-4 and IL-13 target genes [98,99,100,101,102]. The IRS-2/PI3K/AKT/mTOR pathway contributes additional transcriptional modulation mainly through AKT-dependent activation of FoxO1, Bad, TOR, and NF-κB. The Src Homology 2 domain-Containing Phosphatase (SHP)-1, SHP-2 or SH2 Domain–Containing 5′-Inositol Phosphatase (SHIP) pathway is haematopoietically restricted, serving as a phosphatase system to dampen IL-4 and IL-13 signaling [99].

Since IL-4 and IL-13 are anti-inflammatory cytokines, their role in cancer is primarily related to their ability to modulate anti-tumor immune responses. Indeed, IL-4 produced by Th2, and tumor cells is crucial for the expansion of Tumor Associated Macrophages (TAMs). TAMs play a major role in the inhibition of anti-tumor cellular immunity and favor tumor microenvironment modifications, such as the production and activation of myofibroblasts via epithelial–mesenchymal transition, enabling extracellular matrix modification resulting in increased tumor invasiveness [103]. Furthermore, the IL-4/IL-13 axis seems to play an important role in the progression of fibrosis, as it specifically promotes transforming growth factor β1 (TGF-β) expression both in vitro and in vivo in several models of lung, skin, and liver fibrosis [97,104,105].

Focusing on MPNs, contradictory results emerged from two studies assessing plasma IL-4: according to Cacemiro et al., IL-4 levels are increased in all three disease categories as compared to controls, with PMF, in turn, harboring higher levels than PV and ET [22]. Vaidya and co., on the contrary, described similar IL-4 levels across PV, PMF, and controls [20]. IL-13, instead, appears to be nearly absent in healthy subjects while it is significantly increased in the plasma of PMF patients [16]. Interestingly, an increase in IL-13 and TGF1β-producing cells was also detected in BM biopsies of patients diagnosed with myeloid neoplasm with fibrosis (PMF, post-ET MF, and myelodysplastic syndrome with BM fibrosis). Furthermore, an increased number of infiltrating mast cells, the main source of IL-13, was found in MPN patients with a higher grade of BM fibrosis [106].

In line with these findings, Melo-Cardanas et al. recently demonstrated a direct correlation between the IL-13/IL-4 axis and the development of BM fibrosis, using two mouse models of MF: the *JAK2*V617F mutated mice, that gradually develop BM fibrosis 6–8 months after transplantation, and *MPL*W515L mutated mice, showing a more aggressive phenotype with a shorter latency, and a prominent BM fibrosis that arises just 6 weeks after transplantation [21]. 

The levels of 32 cytokines were assessed in the BM and plasma of both mouse models before and after the development of fibrosis. Cytokine level variations were dramatically pronounced in BM but were modest in the plasma, and they were more evident in *MPL*W515L mice as compared to *JAK2*V617F mutated mice, which was likely due to the more aggressive phenotype and faster course of the disease. As BM levels of IL-13 were significantly elevated in both disease models and increased significantly in the advanced fibrotic stage of the disease, the authors further investigated the mechanistic relevance of the IL-13/IL-4/IL-Rα/IL-13Rα1 axis. Higher expression of IL-13Rα1 was found in BM cells from both *JAK*2V617F- and *MPL*W515L-mutated mice as compared to wild types. Specifically, an increased percentage of MKs expressing IL-13Rα1 and IL-4Rα was observed, suggesting that the IL-13/IL-4 axis could be functionally relevant for the aberrant megakaryocytopoiesis that typifies MF. Indeed, in vitro stimulation of murine mutant *MPL*W515L hematopoietic stem cells with IL-13 or IL-4 boosted CD41^+^ MK differentiation. When MK isolated from *MPL*W515L mice were cocultured with wild type mesenchymal stromal cells, the activation of the IL-13/IL-13Rα1/IL-4Rα axis induced a significant increase in fibrosis markers, including α-SMA and collagen 1 and 2. Consistently with the murine model, elevated levels of Il-13 were found in the plasma of MF as compared to ET and PV, and a marked expression of IL-13Rα1 was detected in BM MKs of MF patients as well.

Notably, fostering the IL-13/IL-4/IL-Rα/IL-13Rα1 axis by overexpression of murine IL-13 in the long latency disease model (*JAK2*V617F mice), induced stigmata of disease progression such as increased WBC count, splenomegaly, increased BM TGFβ levels, MKs number, and grading of fibrosis.

Conversely, the IL-13/IL-4/IL-Rα/IL-13Rα1 axis blockage by induced IL-4Rα deficiency in the shorter latency disease model (*MPL*W515L mice) prolonged survival and reduced BM fibrosis, WBC counts, spleen and liver size, and BM MK numbers [21].

Considering that both the IL-4Rα-blocking antibody (dupilumab) and the IL-13-blocking antibody (lebrikizumab) demonstrated efficacy in the treatment of other proinflammatory disorders [107,108,109,110], the blockage of the IL-13/IL-4 axis also presents as a promising anti-fibrotic strategy in the context of MPNs.

## 4. Conclusions and Perspectives

It is increasingly evident that, among myeloid neoplasms, MPNs display a peculiar multifactorial pathophysiology, which reflects phenotypic heterogeneity among disease entities and treatment complexity. This aspect was already crystal clear after the completion of the registrational studies of the first “target therapy” for MPN, i.e., the JAK-inhibitor ruxolitinib [111,112]. In fact, despite the undoubtful clinical benefit, JAK-inhibitors alone do not appear to exert disease-modifying activity toward the most aggressive MPN variant, i.e., myelofibrosis, whilst a combination therapy approach (i.e., ruxolitinib + navitoclax/pelabresib/peginterferon α2a) [113,114,115,116] is emerging as a promising and more effective therapeutic avenue.

Disrupting the vicious chronic inflammatory cycle should be the goal of future investigational treatment strategies in MPNs. Indeed, as elegantly described by Hans Hasselbalch ten years ago, a unique feature of MPN is the detrimental crosstalk between the malignant clone and the chronically inflamed tumor microenvironment (immune, stromal, and endothelial cells), both the source and the target of a plethora of pro-inflammatory mediators and reactive and nitric oxygen species that drive epigenetic changes, genomic instability, and additional DNA mutations, eventually eliciting clonal evolution and disease progression [7]. Cytokines, chemokines, and growth factors have been extensively investigated as key orchestrators of the chronic inflammatory cue in the MPN setting, particularly in MF, and are variably associated with adverse hematological and clinical parameters, aggressive disease behavior, and poor prognosis (reviewed in [33] and Table 2). 

The seven studies described above [14,15,21,23,25,92,94] shifted the focus from the cytokine to its receptor, therefore testing the mechanistic relevance of cytokine/receptor axes. The results are intriguing: not only does MF display—in general—higher cytokine levels but it is also characterized by the overexpression of IL-1, CXCL8, CCL2, and IL-13 receptors on clonal cells, which, by contrast, are virtually absent in normal hematopoietic stem cells. Notably, among them, CCR2 expression is a unique characteristic of MF CD34^+^ cells since the receptor is not expressed by PV and ET, which nevertheless share the same genetic background with MF. Additionally, stimulation of CD34^+^CCR2^+^ from MF cells activates a pro-proliferative and pro-survival signal via Akt phosphorylation [92]. The IL-13 receptor is instead overexpressed by MF megakaryocytic clones, and engagement with IL-13 is capable of inducing STAT6 phosphorylation [21]. 

The selective expression of cytokine receptors (in particular, CXCR1/2, CCR2, and IL-13Rα1) by MF cells suggest that the MF clone is uniquely capable of sensing inflammation, thus suggesting it as a favorite target for the detrimental effects of the inflammatory milieu. In our hypothesis, the intersection of two events—increased cytokine levels and selective expression of their counterpart receptors—creates the “perfect storm” that leads to the end-stage disease in MPNs (Figure 1). 

These very recent data have relevant diagnostic and therapeutic implications. The selective overexpression of cytokine receptors makes them candidate targets for diagnostic purposes. Notably, as surface markers, they are easy-to-test with routine flow cytometry [94]. These studies also pave the way for novel therapeutic strategies in MF based on cytokine receptor antagonists. Indeed, as demonstrated in the above-described studies, cytokine/receptor axis blockage ameliorates the MF phenotype in mouse models of the disease. Moreover, the selective expression of specific receptors by the MF clone, and not by normal cells, leads us to speculate on the potentially limited “off-target” effects of this therapeutic approach by sparing non-clonal cells.

From a pathophysiological standpoint, the burning question now is what are the mechanisms underlying the selective overexpression of cytokine receptors by MF cells? What are the transcriptional/translational/post-translational events that allow only MF cells to expose cytokine receptors on their cellular surface? Does receptor overexpression parallel the progression of ET and PV toward the spent phase? (This latter aspect has indeed been demonstrated for CCR2 [94]).

We believe that current efforts of translational research in MF should be aimed at shedding light on this aspect in order to understand the molecular mechanisms that differentiate MF from other MPN variants.

## Figures and Tables

**Figure 1 biomedicines-11-02462-f001:**
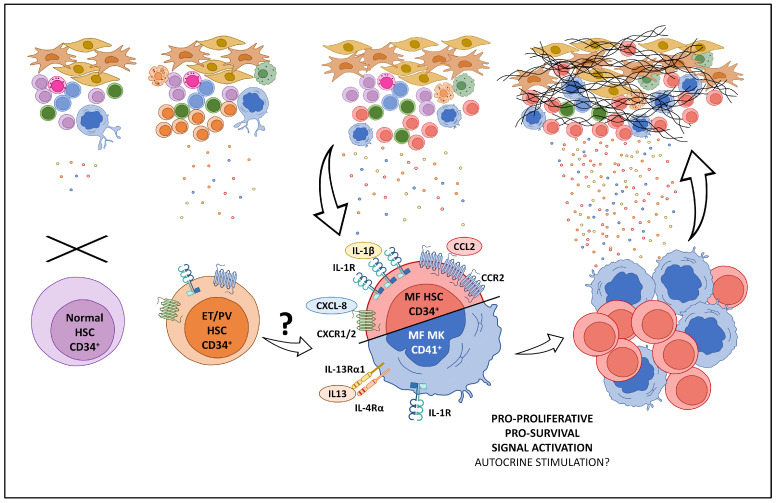
The “perfect storm” that leads to the end-stage disease in MPNs. In normal bone marrow, cytokine levels are low and hematopoietic stem cells do not express cytokine receptors. In PV/ET bone marrow, cytokines levels are high but clonal hematopoietic stem cells express no or low levels of cytokine receptors. In the MF bone marrow, cytokines are high, and the MF clone overexpresses cytokine receptors (IL-1, CXCL8, CCL2, and IL-13 receptors). The activation of these cyto/chemokine axes triggers pro-proliferative and pro-survival signals that boost the clonal expansion and release of additional pro-inflammatory mediators, generating the vicious cycle leading to the “burn out” and “spent phase” with fibrosis, BM anatomical subversion, and high risk of leukemic evolution.

**Table 1 biomedicines-11-02462-t001:** Inflammatory mediator profile in MF.

		MF (vs. HD)	Study Cohort (Number of Patients)	References
Pro-inflammatory cytokines	IL-1α	=	25	[12]
			10	[13]
		↑	34	[14]
			10	[15]
	IL-1β	=	25	[12]
			10	[13]
		↑	127	[16]
			34	[14]
			16	[15]
	IL-2	↑	25	[12]
			10	[13]
			127	[16]
	IL-2R	↑	25	[12]
			127	[16]
			32	[17]
			108	[18]
	IL-6	↑	25	[12]
			127	[16]
			30	[19]
	IL-7	↓ (vs. PV)	127	[20]
	IL-12	↑	127	[16]
	IL-13	↑	20	[21]
	IL-15	↑	127	[16]
			32	[17]
	IL-17	↑	16	[22]
	TNFα	↑	10	[13]
			127	[16]
			16	[22]
	INF-α	↑	127	[20]
			16	[22]
	INF-γ	↓ (vs. PV)	127	[20]
		↑	16	[22]
Anti-inflammatory cytokines	IL-1RA	↑	34	[14]
	IL-4	↑	16	[22]
	IL-10	↑	127	[16]
			16	[22]
	IL-13	↑	20	[21]
Chemokines	MCP-1/CCL2	↑	127	[16]
			32	[17]
		=	16	[22]
	MIP-1α	↑	127	[16]
		↓	16	[22]
	MIP-1β	↑	127	[16]
	IL-8	↑	127	[16]
			32	[17]
			35	[23]
	RANTES	↑ (vs. PV)	127	[20]
		=	16	[22]
	IP-10	↑	127	[16]
			16	[22]
	MIG	↓ (vs. PV)	127	[20]
		↑	127	[16]
	CCL11	↓ (vs. PV)	127	[20]
Growth factors	GM-CSF	↑	16	[22]
		↓ (vs. PV)	127	[20]
	G-CSF	↑	127	[16]
	HGF	↑	127	[16]
	VEGF	↑	127	[16]
		↓ (vs. PV)	127	[20]
	EGF	↑ (vs. PV)	127	[20]
	FGF	↑ (vs. PV)	127	[20]
	TPO	↑	25	[12]
	TGF-β	↑	10	[24]

Summary of deregulated cytokine levels in peripheral blood and bone marrow of myelofibrosis (MF) patients (by ELISA, cytokine array) as compared to control healthy subjects (HD) or polycythemia vera (PV) where indicated. Cytokines are grouped according to their function. ↑ increased; = similar; ↓ reduced.

**Table 2 biomedicines-11-02462-t002:** Function and impact on MF phenotype of inflammatory mediators in MF.

Cytokine/Chemokine/ Growth Factor	Function	MF Phenotype	Refs.
HGF	Mitogen, motogen, and morphogen for a variety of epithelial cells.	Splenomegaly, leukocytosis, association with *JAK2*V617F.	[16]
IL-1α	Pro-inflammatory, target of inflammasome.	Association with *JAK2*V617F, BM angiogenesis. IL-1/IL-1R blockage ameliorates fibrosis in mouse models.	[12,14,16]
IL-1β	Pro-inflammatory, target of inflammasome.	Association with *JAK2*V617F. IL-1/IL-1R blockage ameliorates fibrosis in mouse models.	[14,15,16]
IL-2	Survival, proliferation, differentiation, and function of T lymphocyte subsets and NK cells.	Association with *JAK2*V617F, transfusion dependency, hepatosplenomegaly, and BM angiogenesis.	[12,13,16]
sIL-2R	Repressing/supporting immunity via interaction with IL-2.	Reduced OS *, RBC transfusion dependency, leukocytosis, association with *JAK2*V617F, hepatosplenomegaly, and BM angiogenesis.	[12,16,17,18]
IL-4	Anti-inflammatory, allergy, airway hyperresponsiveness, tissue eosinophilia, mastocytosis, IgE Ab production, and fibrosis.	IL-4R blockage reduces fibrosis in mouse models.	[21,22]
IL-6	Synthesis of acute phase proteins in liver, Ab production, differentiation of naïve CD4^+^ T cells into effector T cells, activation of vascular endothelial cells, HSC Differentiation, and MK maturation.	Constitutional symptoms, association with *JAK2*V617F, reduced PLT count, hepatosplenomegaly, BM angiogenesis and degree of fibrosis, circulating CD34^+^ cells, anemia, activation of inflammasome.	[12,16,19]
IL-8	Neutrophil chemotaxis and activation.	Reduced OS *, reduced LFS, RBC transfusion dependency, constitutional symptoms, male gender, degree of BM fibrosis, presence of circulating blasts, and leukocytosis. CXCR1/2 blockage reduces fibrosis in mouse models.	[17,23,25]
IL-10	Anti-inflammatory, limits secretion of pro-inflammatory cytokines, deactivation of macrophages, and inhibition of T cell proliferation.	Reduced RBC and RBC transfusion dependency.	[16,22]
IL-12	Promotes generation of pro-inflammatory Th1 and Th17 cells.	Reduced OS *, association with *JAK2*V617F., unfavorable karyotype, and RBC transfusion dependency.	[16]
IL-13	Anti-inflammatory, allergy, airway hyperresponsiveness, tissue eosinophilia, mastocytosis, IgE Ab production, and fibrosis.	Degree of BM fibrosis. IL-13 overexpression in mouse models promotes fibrotic phenotype.	[21]
IL-15	Survival, proliferation, and activation of natural killer (NK) and CD8^+^ T cells.	Reduced OS *, male gender, and splenomegaly.	[16,17]
IL-17	Pro-inflammatory, promotes activation of endothelial cells and monocytes.	Reduced RBC count.	[22]
INF-α	Activation of innate immune response and NK cells, inflammasome activation.	Increased PLT count.	[22]
IP-10	Chemoattractant for activated T and NK cells, and fibroblast activation.	Reduced OS, reduced PLT count, leukocytosis, older age, and association with *JAK2*V617F.	[16,22]
MCP-1/CCL2	Monocyte chemotaxis and activation, cell proliferation, and fibroblast activation.	Degree of BM fibrosis and RBC transfusion dependency.	[16,17]
MIG	Chemoattractant for activated T and NK cells.	Association with *JAK2*V617F and male gender.	[16]
MIP-1α	Monocyte chemotaxis and activation, and activation of Th1 response.	Reduced Hb and RBC, RBC transfusion dependency, and male gender.	[16,22]
MIP-1β	Monocyte chemotaxis and activation, activation of Th1 response	Reduced OS and reduced PLT count.	[16]
RANTES	Trafficking and homing of T cells and monocyte, basophil, eosinophil, NK cell, dendritic cell, and mast cell activation.	Increased PLT count.	[22]
TGF-β	Promotes fibroblast growth and activation, inhibits tumor development at early stages and drives tumorigenesis at later stages.	Degree of BM fibrosis and depletion of the pool of normal HSCs. TGF-β blockage reduce fibrosis in mouse models.	[26,27]

* Also in multivariate analysis including DIPP-plus risk stratification. Abbreviations: BM = bone marrow; DIPP = The Dynamic International Prognostic Scoring System; LFS = leukemia free survival; Hb = hemoglobin; HSCs = hematopoietic stem cells; n/d = not determined; OS = overall survival; PLT = platelets; and RBC = red blood cells.

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
