# Peer review of "Functionally Relevant Cytokine/Receptor Axes in Myelofibrosis"

_biomedicines, 2023, doi:10.3390/biomedicines11092462_

Round 1

Reviewer 1 Report

Dysregulation of inflammatory signaling is known to be a key feature of myeloproliferative neoplasms (MPN), especially myelofibrosis (MF). Indeed, MF is considered the prototype of onco-inflammatory hematological cancer. Elevated levels of circulatory and bone marrow cytokines are a well established feature of all MPNs. However, the review authors reviewed the major cytokine systems especially activated in MF and how cytokine receptor targeting is emerging as a new therapeutic avenue in this disease.

In general, the review is quite interesting, covers the topic well, is logically and consistently structured. However, it seems to me that a summary table should be added for MF studies with a description of the number of patients in each study, the list of detectable cytokines, a description of the dynamics of cytokines, etc. Without this, it is not clear how developed the topic is and whether the authors of various studies show a reproducible result.

Author Response

We thank the Referee for the comment.

As suggested, we added a new table (New Table 1), summarizing the landmark studies assessing levels of cytokines/chemokines and growth factors in MF. The dynamic of cytokines as compared to healthy subjects or other MPNs, and the patient cohort for each study is reported. We also specified that New Table 2 (previously Table 1) focuses on those mediators for which an impact on MF disease phenotype has been reported.

Reviewer 2 Report

Comments:

    The manuscript describes " Is cytokine receptor overexpression the hallmark of myelofibrosis? ". Myelofibrosis is a mysterious myeloproliferative neoplasm that, in addition to the extracellular matrix, also secretes a large number of pro-inflammatory cytokines, growth factors, and components (laminin and collagen). Subsequent disruption of the interplay between megakaryocytes, osteoblasts, endothelial cells, stromal cells, and myofibroblasts in the bone marrow eventually leads to the development of fibrosis and osteosclerosis. Disease-modifying drugs and an understanding of the pathogenesis of myelofibrosis are still lacking., but several points need to be clarified.

Comment:

1. The article title is inappropriate. 

2. The pathogenesis of myelofibrosis needs to be described in detail and the description of mechanistic research needs to be deeper.

3. Current myelofibrosis disease-modifying drugs and mechanisms need to be described in detail and drawn to be added to the table.

Moderate editing of English language required

Author Response

We thank the Referee for the comments.

Point n. 1. As suggested, we modified the title of the manuscript as follows: “Functionally relevant cytokine/receptor axes in myelofibrosis”. We hope this title will work better.

Points n. 2 and 3. We believe that a detailed description of the pathogenesis of myelofibrosis, its therapy, and the mechanisms of action of disease modifying drugs, although of undoubtful interest, are out of the scope of this review and out of the scope of the topic of this Special Issue.

Moreover, all these aspects have been recently covered by other Experts in the field (PMID: 37584267, PMID: 37002477).

Of course we agree with the Referee that our understanding of the mechanistic events that lead to MF is still fragmentary. Given the topic of this Special Issue, we decided to focus on the role of inflammation, discussing the new data that derive from recent studies which demonstrate not only that cytokines are increased in MF (as in MPN in general), but also – and more importantly - that MF hematopoietic stem cells can uniquely overexpress cytokine receptors, thus becoming selective targets of the inflammatory milieu.

Round 2

Reviewer 1 Report

I have no more comments on the article. I think that in its present form the article can be recommended for publication.

Reviewer 2 Report

accepted

Minor editing of English language required